# The mechanical influence of densification on epithelial architecture

Christian Cammarota[1], Nicole S. Dawney[2], Philip M. Bellomio[2], Maren Jüng [2], Alexander G. Fletcher[3], Tara M. Finegan [2¤]*, Dan T. Bergstralh [1,2,4¤]*

**1** Department of Physics and Astronomy, University of Rochester, Rochester, New York, United States of America, **2** Department of Biology, University of Rochester, Rochester, New York, United States of America, **3** School of Mathematics and Statistics, University of Sheffield, Sheffield, United Kingdom, **4** Department of Biomedical Genetics, University of Rochester Medical Center, Rochester, New York, United States of America

¤ Current address: Division of Biological Sciences, University of Missouri, Columbia, Missouri, United States of America

* tara.finegan@missouri.edu (TMF); dan.bergstralh@missouri.edu (DTB)

**Data Availability Statement:** The computational code can be accessed at our lab's Github page: https://github.com/Bergstralh-Lab.

**Funding:** This work was supported by an National Science Foundation CAREER award, 2042280 (PI:

## Abstract

Epithelial tissues are the most abundant tissue type in animals, lining body cavities and generating compartment barriers. The function of a monolayered epithelial tissue–whether protective, secretory, absorptive, or filtrative–relies on the side-by-side arrangement of its component cells. The mechanical parameters that determine the shape of epithelial cells in the apical-basal plane are not well-understood. Epithelial tissue architecture in culture is intimately connected to cell density, and cultured layers transition between architectures as they proliferate. This prompted us to ask to what extent epithelial architecture emerges from two mechanical considerations: A) the constraints of densification and B) cell-cell adhesion, a hallmark feature of epithelial cells. To address these questions, we developed a novel polyline cell-based computational model and used it to make theoretical predictions about epithelial architecture upon changes to density and cell-cell adhesion. We tested these predictions using cultured cell experiments. Our results show that the appearance of extended lateral cell-cell borders in culture arises as a consequence of crowding–independent of cell-cell adhesion. However, cadherin-mediated cell-cell adhesion is associated with a novel architectural transition. Our results suggest that this transition represents the initial appearance of a distinctive epithelial architecture. Together our work reveals the distinct mechanical roles of densification and adhesion to epithelial layer formation and provides a novel theoretical framework to understand the less well-studied apical-basal plane of epithelial tissues.

## Author summary

Epithelial tissues have critical functions in animal bodies–including protection, secretion, absorption, and filtration. To perform these functions, the component cells that make up the tissue must maintain their shape and organization. Loss of epithelial tissue organization leads to disease such as cancerous carcinoma. In this study, we explored the

DTB), National Institute of Health Grant R01GM125839 (PI: DTB), and the UK's Engineering and Physical Sciences Research Council (grant EP/W024144/1 to AGF). The funders did not play any role in the study design, data collection and analysis, decision to publish, or preparation of the manuscript.

**Competing interests:** The authors have declared that no competing interests exist.

biophysical factors that drive epithelial shape using a combination of computational modelling and experimental studies. As cultured cells proliferate and consequently densify, they undergo a series of developmental transitions before achieving a mature architecture. Given the relationship between architecture and cell density, we asked to what extent cell crowding alone can explain the developmental transitions observed. We find that while crowding is sufficient to explain the initial feature of architecture development, namely the appearance of cell height, subsequent transitions also rely on cell-cell adhesion, a hallmark feature of epithelia.

## Introduction

Epithelial tissues are cohesive cell sheets that serve to mechanically protect and segregate animal body compartments. The most common epithelial architecture is a "simple" monolayer (a sheet of cells that is one cell thick), with component cells sharing a common apical-basal polarity axis, adhering closely to their lateral neighbors, and demonstrating a uniform distribution of nuclei with respect to the apical-basal axis [1,2]. Epithelial tissue function relies on tissue architecture, and a disruption of this architecture can lead to disease. For example, the most common form of adult cancer is solid carcinoma, which results from epithelial overproliferation and subsequent architectural disruption [3].

Epithelial cell shapes have predominantly been studied in the apical plane, largely due to historical technical limitations of light microscopy. While the molecular effectors of apical-basal polarity and the relative positions of cell-cell junctional complexes have been widely studied, the physical basis for cell shape in the apical-basal axis of epithelial tissues remains relatively unexplored. In particular, it is unclear how the position of adhesion complexes and the magnitude of adhesive forces that these complexes confer contribute to epithelial architecture.

Across metazoans, epithelial cells in a proliferating tissue are chiefly hexagonal at the apical surface [4–6]. One explanation for this observation is that hexagonal packing represents a minimum energy state [4,7]. Even in the absence of minimal packing, cell division pressure can give rise to hexagonally packed tissue as a geometric consequence of Euler's theorem [6]. Because cell shapes and cell packing are intertwined, cell shape regularity (as measured by circularity) can be used to reflect packing efficiency. We previously described that the apical cell shapes of confluent cultured epithelial cells become more regular as they proliferate and the cultured tissue densifies [8] (S1 Fig). Cell shapes in the plane of the tissue are widely studied in the context of understanding and predicting the rheological properties of epithelial tissues. Vertex models offer a simple theoretical framework that has proven useful for elucidating and predicting the topology and dynamics of epithelial tissues within the tissue plane [9,10]. Vertex models have been used successfully to explore the mechanical basis for jamming/unjamming transitions in experimentally measured tissue systems [11–14]. For tissues undergoing significant active cell rearrangements, such as the *Drosophila* germband, more geometrically detailed mechanical models have been proposed to explain intercalation dynamics and cell shearing, including those that go beyond the vertex model's approximation of cell-cell contacts as single edges [15].

In contrast, the connection between cell shapes within the tissue plane and the observed developmental progression of epithelialization in the apical-basal axis remains poorly understood, due in part to limitations of existing computational models. While several studies have used computational modelling to explore different mechanical contributions to the deformation of epithelial sheets in this axis, these have focused primarily on gastrulation and similar

processes, neglecting cell-substrate interactions. Such studies include the pioneering work of Odell et al [16] who modelled an embryo cross-section as a ring of viscoelastic cells with inter-connected vertices with active contractility in response to stretching. This work, and more recent modelling efforts [17–21] have recapitulated essential features of *Drosophila* ventral fur-row formation. Other applications of such modelling have included, for example, an identification of a two-step mechanism underlying early ascidian gastrulation [22]. These models capture key features of epithelial cell mechanics in the transverse dimension, including active remodeling of cytoskeletal components and heterogeneous adhesion, but tend to represent both apposed cortices in each bicellular junction by a single edge, preventing neighboring cells from coupling/uncoupling to/from one another. In particular, previous models have not addressed epithelial shape development *ab initio*, nor how fundamental mechanical factors drive epithelial layer architecture diversity.

An unbiased image analysis pipeline called ALAn (Automated Layer Analysis) was recently developed to measure apical-basal architecture in Madin-Darby Canine Kidney (MDCK) cul-tured epithelial cell layers [8]. ALAn recognizes a developmental series of three organized architectures (Immature, Intermediate, and Mature), and a fourth category of exclusion (Dis-organized) based on the average shape and nuclear position of component cells in the apical-basal plane of the tissue [8]. Organized architectures are distinguished as follows: 1) Immature layers are composed of flat cells which do not form significant cell-cell contacts; 2) Intermedi-ate layers are taller and have well-defined cell-cell contacts; 3) Mature layers are the tallest organized layers, with well-defined cell-cell contacts and a flattened apical surface (S1 Fig). Cultured layers transition through the developmental series of architectures as density increases, with each architecture occupying a particular density regime (S1 Fig).

In this study we used a combination of computational modeling and cultured-cell experi-mentation to ask whether the developmental transitions from one organized architecture to the next are mechanically driven. Given that the mechanical parameter most obviously associ-ated with these transitions is cell density, we focused on whether these transitions are driven by cell crowding. We find that crowding can explain the first transition (Immature to Interme-diate) but not the second, and that the initial appearance of Intermediate architecture is not restricted to epithelial cells. However, we also identify two distinct morphological categories within the Intermediate architecture. Our findings indicate a new transition in confluent cell architecture within the Intermediate category that is restricted to epithelial cells.

## Results

### Development of a 2D polyline cell-based computational model

Computational models, in particular vertex models, have been widely used to quantitatively study and predict epithelial architecture and dynamics in the plane of the tissue [10,23]. The shape of cells in the apical-basal axis during the earliest stages of epithelial layer formation are not well-described by simple polygons, and vertex models don't well-predict cell shapes at low densities. We therefore decided to develop a new computational approach to systematically probe how cell density, cell-substrate, and cell-cell adhesion contribute to epithelial architec-ture. We developed a novel 2D polyline computational model of cell shape dynamics in the apical-basal plane, extending previous models that allow for arbitrarily curved and irregular cell shapes [24,25].

We modeled cells growing on a fixed, rigid substrate, representing the basement membrane for epithelia *in vivo*, and a plastic or glass dish coated with extracellular matrix *in culture*. The XZ cross section (i.e. apical-basal axis) of each cell's cortex is modelled by a polygon compris-ing many vertices or 'nodes'; similarly, the substrate is modelled by a fixed line (rigid non-

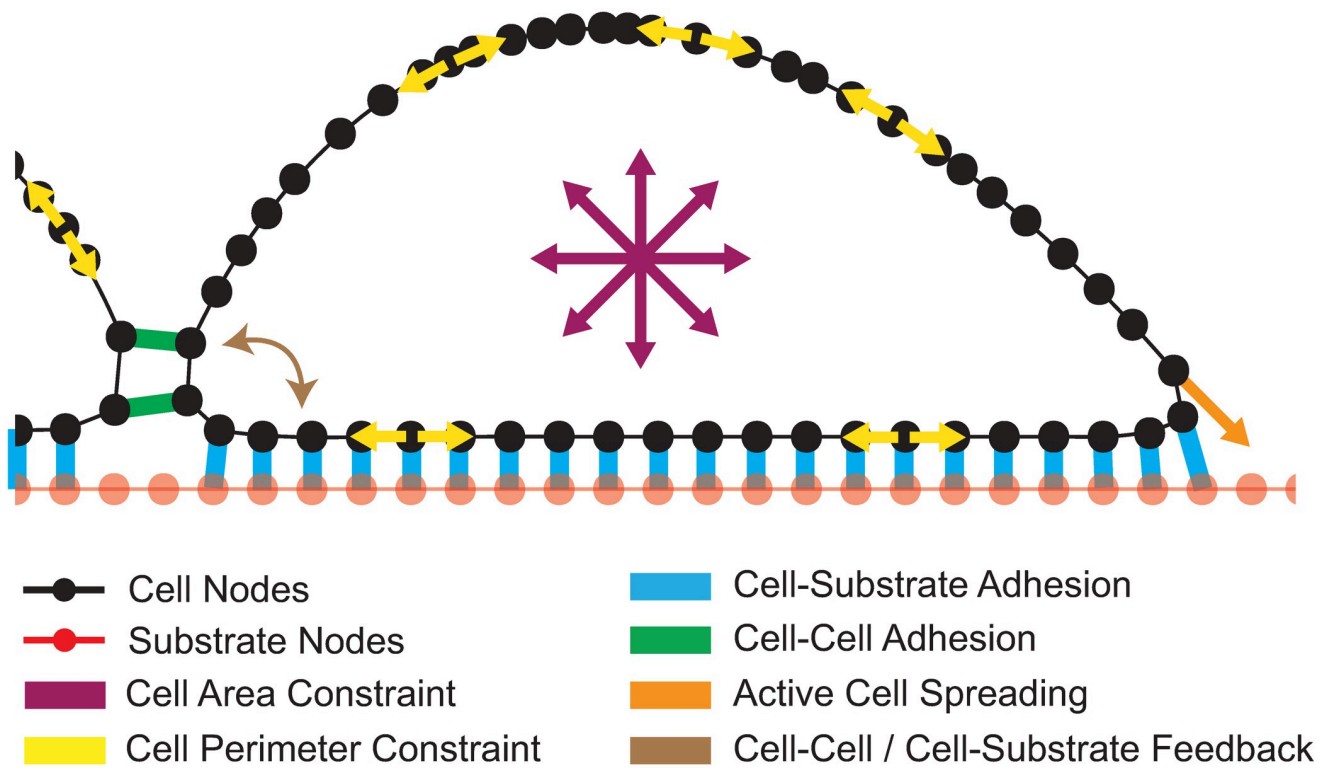

**Fig 1. Introduction to the computational cell model.** Schematic representing the forces accounted for in the computational model.

deformable body) discretized into a large number of nodes (Fig 1). The spacing between substrate nodes is set to the initial cell node spacing. Each cell experiences the following forces: external adhesions formed by cell-cell and cell-substrate contacts (formed by pairs of nearby cell-cell nodes and cell-substrate nodes, respectively); internally generated forces regulating size and shape via cytoskeletal dynamics; and an active spreading force driven by actin-based lamellipodia (Fig 1). While cell-cell interfaces are typically vertical in our model, this is an emergent feature of the parameters rather than an implemented requirement.

In addition to these forces, a feedback mechanism models the crosstalk between cell-substrate and cell-cell adhesions [26], allowing for cells to grow their cell-cell contacts and find neighbors at an increased range after cell-cell contacts initiate. This feedback mechanism is predicated on reported work showing that the initiation of cell-cell adhesion causes A) cortical remodeling at cell-cell contacts [27,28], B) elongation of cell shape [29] and C) increased polymerization of cortical actin [27]. In agreement with the latter point, we observe increased protrusions in MDCK cells that contact a neighbor (S1 Movie).

## Implementation of active spreading in the model

We first attempted to model the behavior of a single cell in contact with a substrate. Our initial expectation was that cell-substrate adhesion would be sufficient to promote the flat shape of isolated cells cultured on a substrate (S2A Fig), but we found this not to be the case; simulated cells maintain a round shape under the influence of substrate adhesion alone (S2B Fig). In culture, spreading relies on active remodeling (protrusion) of the cytoskeleton in addition to adhesion [30–32]. We therefore implemented an active cell-spreading component to our

simulations, introducing a constant force (1.6e-5 N), pointing 45˚ outward and below horizontal that acts on nodes immediately adjacent to the outermost substrate connections. Active spreading initiates once 10% of all nodes contact the substrate to account for the passive spreading phase of cells on substrates. The addition of an active spreading force qualitatively recapitulated the cell shapes observed in culture (S2C Fig).

We also incorporated contact inhibition of spreading, a phenomenon similar to contact inhibition of locomotion, into our model [33]. To achieve this, the spreading force drops to zero when a simulated cell is in contact with a neighbor. Contact inhibition of spreading is implemented at each edge of the cell independently, meaning that spreading continues on the other side of the cell until/unless it also contacts a neighbor.

The refined model recapitulates our live cell imaging. Based on our observations, both our cultured and modeled cells tend to maximize substrate connections and will preferentially make cell-substrate contacts instead of cell-cell contacts (S1 Movie). A representative simulation of two cells in proximity closely reflects the same configuration of cultured MDCK cells (Fig 2A).

## Intermediate epithelial architecture arises as simulated cells densify

Cultured monolayer architecture changes as the layer densifies (S1 Fig) [8], suggesting the possibility that epithelial architecture arises from cell crowding. We used our computational model to test this possibility, varying cell density by limiting the amount of available basal substrate. As a 2D model of a 3D system, density cannot be directly compared between the simulated and physical layers. We therefore report simulated cell density in terms of arbitrary units (Arb.), derived by scaling the linear cell density. At essentially unlimited substrate, cells spread without developing many contacts and cells form architectures resembling Immature architecture (Fig 2B). Lateral surfaces develop as the available substrate is decreased (Fig 2B'). These surfaces are observed in our simulations when more than one cell-cell connection is made between cells. Given the average number of nodes in each cell, the transition from an Immature to an Intermediate architecture will happen when ~10% of simulated cell surface area are participating in cell-cell adhesion.

The flat apical surface characteristic of Mature layers is not recapitulated in our simulations even at the highest densities. We confirmed this by comparing the ratio of the lengths of the arced apical surface to basal surface in MDCK cells and simulated cells. In both cases, this ratio increases gradually over the densities associated with Immature and Intermediate architectures. In the simulated cells, a gradual decline begins at a density near 7 Arb (S3 Fig). In MDCK cells the ratio demonstrates a steeper decline beginning at ~$7\times10^3$ cells/mm$^2$, correlating with the appearance of flattened apices (S3 Fig). We expect that flattening is likely related to molecular asymmetries at the subcellular level (along the apical-basal cell axis) that promote apical contractility. Implementation of these asymmetries into a revised model is a goal for future work.

Our simulations recapitulate the development of Immature and Intermediate, but not Mature, architectures. These results suggest that the appearance of Intermediate architecture is a straightforward mechanical consequence of deformable objects crowding together. In contrast, the transition from Intermediate to Mature architecture relies on an additional biophysical step(s) initiated by densification.

## Division pressure drives the immature to intermediate transition

Our modeling and culture data predict that cell density is the driving force behind the emergence of Intermediate epithelial architecture. This raises the question of what drives

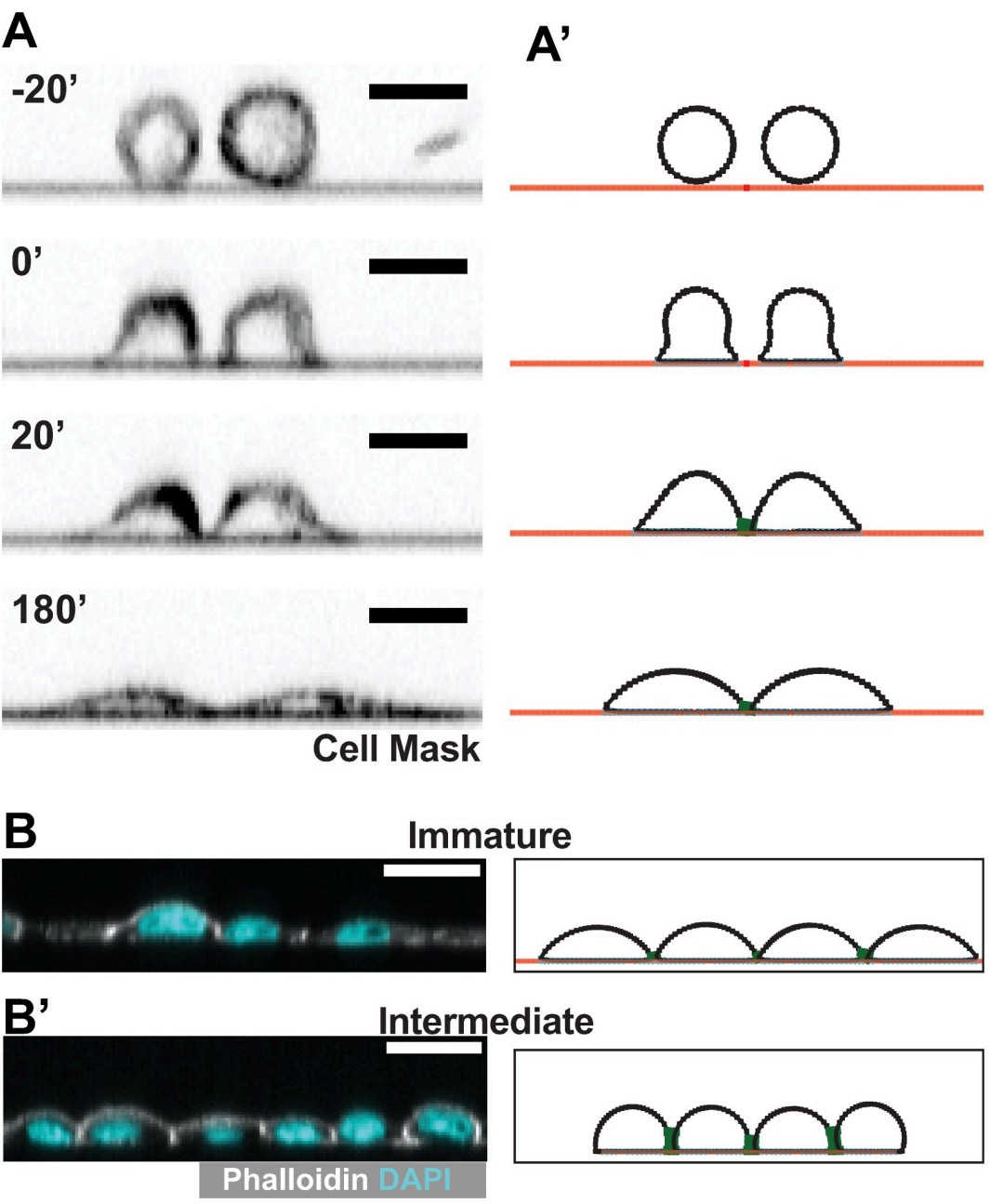

**Fig 2. The computational model recapitulates the behavior of MDCK cells in culture.** Side by side comparison of live MDCK cells (left) and simulated 'cells' (right) show that the model can recapitulate real cell behavior in the contexts of: A) cell 'plating'- dropping cells onto a substrate, (Full simulations shown in S2 and S3 Movies) and B) layer densification, and therefore epithelialization (Full simulations shown in S4 and S5 Movies). Simulated cells exhibit Immature architecture at low densities, and Intermediate architectures at higher densities. Scale bars = 20 μm.

densification in epithelia. The most obvious answer is cell division (other routes of confinement, and therefore densification, could be the application of extrinsic compressive forces, or collision with other cell colonies). We therefore implemented division in our computational model (described in *Materials and Methods*).

We simulated growth of a three-cell colony along an unbound substrate, "seeding" it such that a central cell divides before its neighbors. After this cell divides, the two daughter cells develop multiple cell-cell contacts with each other and their neighbors, reflecting Intermediate architecture (Fig 3A). Given additional time, simulated cells spread along the substrate and return to an immature architecture, but the timescale for this rearrangement is comparable to the cell cycle. In agreement with earlier work which predicts that a proliferative colony will

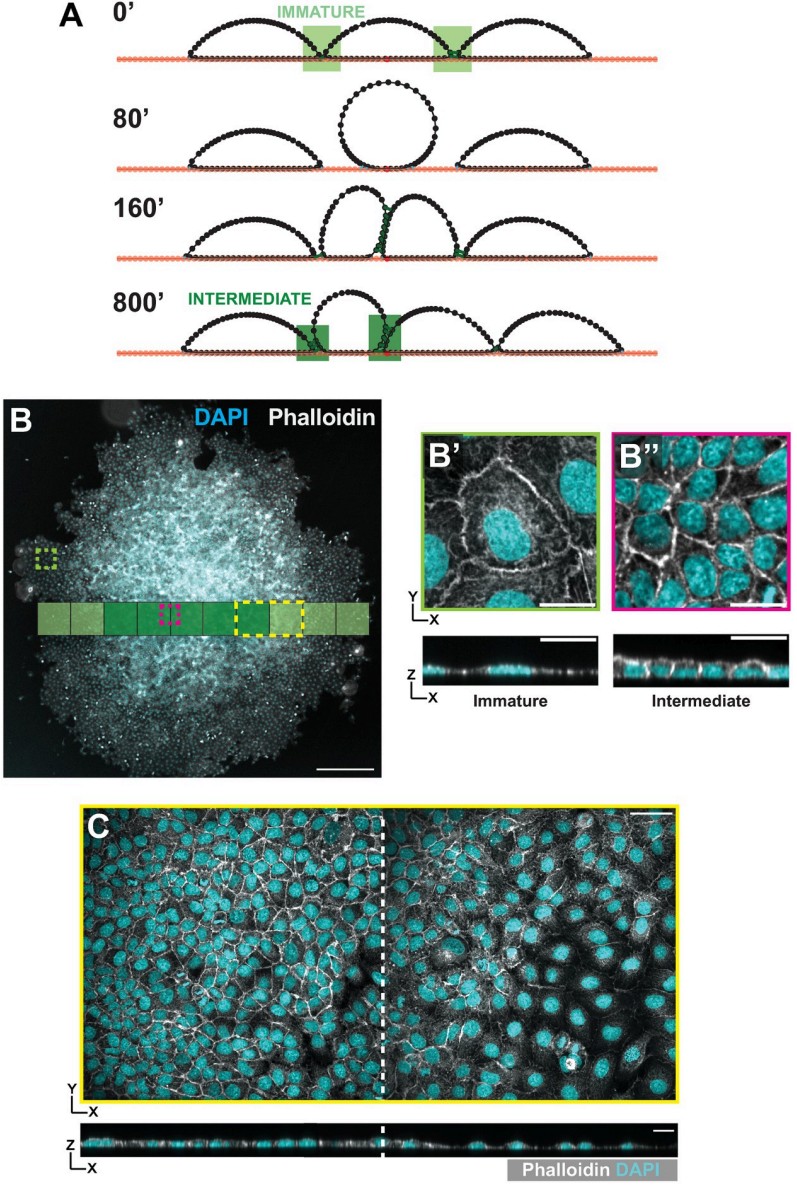

**Fig 3. Division pressure drives the Immature to Intermediate architectural transition.** A) Timepoints from a simulation of a 3-cell layer with a division implemented in the central cell. Division leads to an increase in cell-cell density at the center of the monolayer, and a gradient of cell-cell contacts from the edge to the center of the layer. Full simulation shown in S6 Movie. B) An isolated cell colony of MDCK cells grown from sparse cell seeding onto the culture substrate. The colony exhibits a density gradient from the center to the edges as predicted by the model. Cells at the edge of the colony form an Immature architecture (B, and B') whereas cells in the dense central region form an Intermediate architecture (B and B"). C) High resolution imaging of the region highlighted in the yellow box from B shows the gradient in cell density. Scale bar in B = 500 μm. All other scale bars = 20 μm.

exhibit a cell density gradient with its highest point at the center, the colony edge expands outward through a combination of proliferation throughout the colony and active spreading by outer cells [33–35].

Both theory and our simulations predict that in combination with active spreading, division pressure promotes the transition from Immature to Intermediate architecture [36]. We tested this prediction by culturing MDCK cells at a low density, creating colonies of cells that were smaller than the growth area of our culture plates and therefore unconfined at their edges (Fig 3B). Our MDCK colonies clearly showed a density gradient from the center to the edge of the colonies as predicted (S4 Fig). Using our image analysis pipeline (ALAn) [8], we found that layer architectures are Immature at the colony edge and Intermediate at the colony center (Fig 3B and 3C). These results agree with previous studies showing that cells in the central region of a growing colony demonstrate greater shape regularity (with respect to the tissue surface) than cells at the edge, as expected from the transition from Immature to Intermediate architecture [36–38]. Together these findings suggest that densification is sufficient to drive the transition from Immature to Intermediate architecture.

## Intermediate architectures develop when cell-substrate adhesion and spreading are impaired

Having determined that Intermediate architecture development relies fundamentally on crowding, we next set out to investigate the role played by cell-substrate adhesion. *In vivo*, epithelia are anchored in position by interaction with a basement membrane rich in extracellular matrix (ECM) [39]. The basal substrate provides physical and biological networks that regulate cell behavior and tissue architecture [40]. This is modeled in culture by providing ECM on the cell culture substrate and is necessary for epithelialization [41].

To model the development of Intermediate architecture *in silico*, we time-evolved a simulated four-cell colony for a period corresponding to 24 hours in culture. We restricted our modeling to densities below 7 Arb (S3 Fig). We performed simulations over a range of cell-substrate adhesion values, starting with 20 N/m, which is physiologically relevant and used as our standard [42]. The relationship between cell-substrate adhesion (as mediated by matrix receptors) and active protrusion is difficult to parse experimentally, but we and others observe that limited MDCK cell spreading occurs even on uncoated glass [43] (S5A Fig). We therefore undertook simulations using three models for active spreading: 1) the spreading force is held constant at all substrate adhesions; 2) the spreading force scales linearly with substrate adhesion; and 3) the spreading force scales non-linearly with substrate adhesion. We consider the first condition to represent an upper bound and the second to represent the lower bound, whereas the third most closely represents our observations in culture.

We used the proportion of total interaction nodes (per cell) that participate in cell-substrate or cell-cell adhesion as proxies for cell-substrate interface and cell-cell interface length. These parameters normally decrease and increase (respectively) as an MDCK layer matures (S5B and S5C Fig).

As expected, we found that reducing cell-substrate adhesion strength reduces the number of cell-substrate connections in all three spreading conditions (the non-linear scaling force is shown in Fig 4A and 4B, the constant force in S5D and S5E Fig, and the linear scaling force in S5F and S5E Fig). Somewhat counterintuitively, it also reduces the number of cell-cell connections, especially at higher densities. This is because the shape of a simulated cell in our model is determined largely by a balance between two parameters, spreading and the internal regulation of cell shape, which is buffered by the positive feedback between cell-cell and cell-substrate

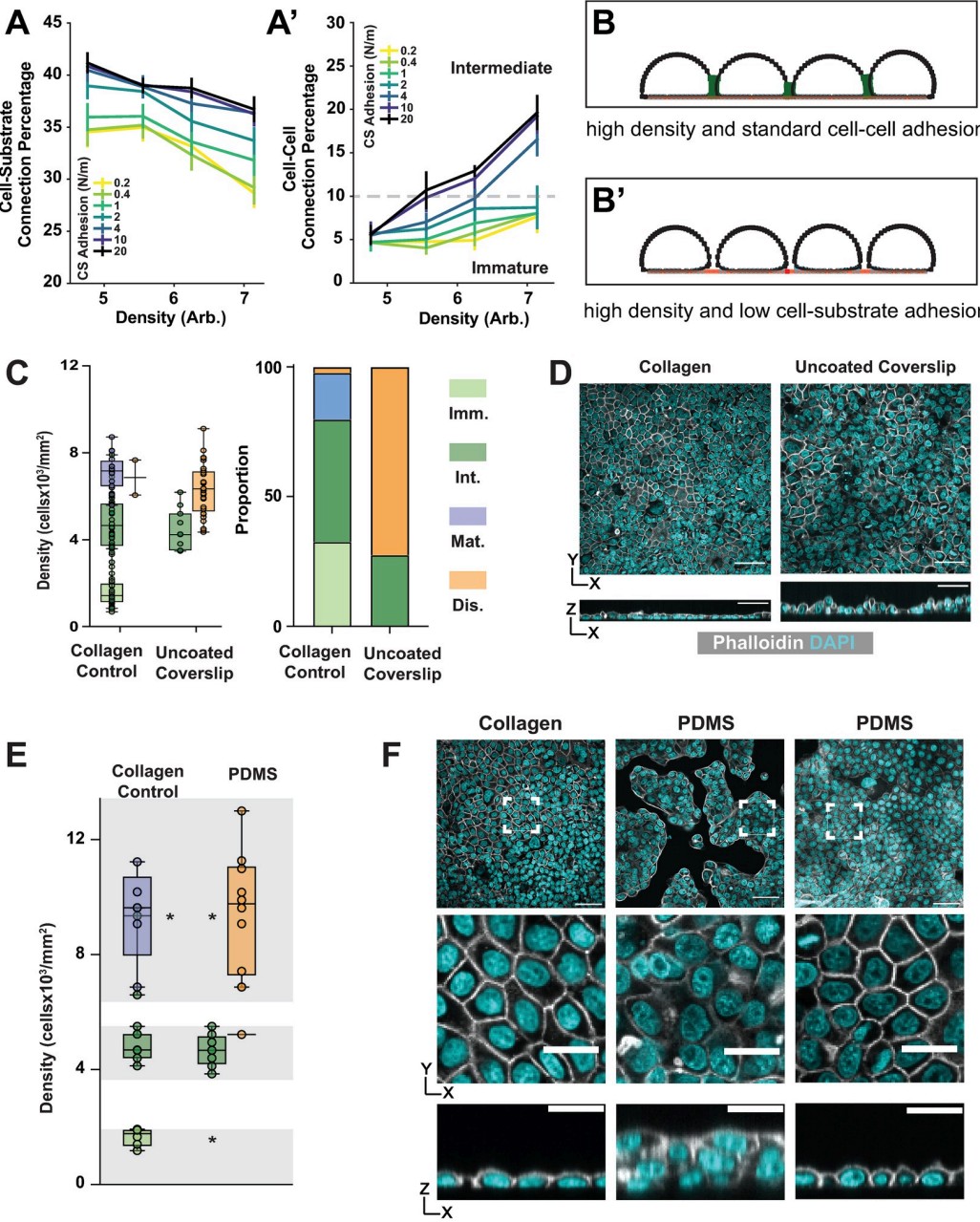

**Fig 4. Intermediate architecture can form despite decreased cell-substrate adhesion.** A) Cell-substrate connections decrease (A) and cell-cell connections increase (A') as cell density increases across tested values of cell-substrate adhesion strengths when spreading is scaled nonlinearly in simulations. The dashed line represents the connection percentage required for Intermediate architectures to arise in the model (A'). B) Model representative displaying the final result from decreasing cell-substrate adhesion strengths. C and D) Plating cells on uncoated coverslips leads to primarily Disorganized architectures, but Intermediate architectures can still be found. Representative images shown in D. Chi-squared test *p*-value < 0.0001. Scale bar = 50 μm. E and F) Plating cells on a PDMS membrane leads to islands of primarily Disorganized architecture but Intermediate architectures are still found (E). Representative images shown in F. Highlighted areas in E represent density cut-offs at which different architectures should be observed. Due to the presence of islands, smaller areas of each image are analyzed (dashed boxes, F) at the densities of which each layer type should be observed.

adhesion described above. Simulated cells become more circular (Figs 4B, S5E and S5G) when this balance is shifted by weakened cell-substrate adhesion.

Spreading strength modulates the role of substrate adhesion in the transition from Immature to Intermediate architecture in our simulations. In both the nonlinearly scaled and constant strength spreading conditions, Intermediate architectures develop even when cell-substrate adhesion is reduced by as much as 80% (from 20 N/m to 4 N/m) (Figs 4A' and S5D'). In the linearly scaled spreading condition, Intermediate architectures are observed at 50% adhesion strength (S5F' Fig). Thus, while Intermediate architectures develop in each spreading mode, the amount of substrate adhesion required varies inversely with the spreading strength.

Our simulations show that Intermediate architectures can develop when spreading and substrate adhesion are reduced. We tested this in culture by changing the substrate. The standard culture wells used in this and our earlier study are coated with collagen, an ECM component that provides a basis for receptor-mediated adhesion but is not required for MDCK cell adherence to the culture well, meaning that adhesion is diminished but not lost in its absence [43]. We shifted to uncoated wells and found that the layer architecture profile changed dramatically:

Disorganized layers predominated at the densities we expect to find Mature layers ($> 5\text{x}10^3$ cells/mm$^2$) and no Immature architectures were observed. (Fig 4C and 4D). Both observations are predicted by our conceptual model for monolayer development, which is that monolayers rely on a competition between cell-cell and cell-substrate adhesion. In the collagen-free condition, the balance is expected to shift towards cell-cell adhesion, meaning that cells will tend to accumulate (clump) rather than cover the substrate. At high densities this shift causes cells to pile up, leading to Disorganization. At low densities it prevents confluence. In agreement with the latter, we found that approximately 20% of regions imaged on the uncoated substrate were subconfluent at 24 hrs, by which time cells on the collagen substrate had reached confluence.

Roughly 30% of the layers analyzed demonstrated the characteristics of Intermediate architecture, both with respect to apical-basal and apical (surface) cell shapes. These layers also fell in the expected ranges for cell density (Fig 4C) and cell shape regularity with respect to the surface (S5H Fig). This result agrees with our simulations; Intermediate architectures can develop despite reduced cell-substrate adhesion.

We also plated cells on a biologically inert polydimethylsiloxane (PDMS) membrane. This substrate not only lacks collagen, as in the previous experiment, but is also less stiff (~1 MPa for our membrane from ~1Gpa for our coverslip) and should therefore curtail active spreading [44–46]. In agreement, cells plated on PDMS do not spread to cover the available substrate, as they do on collagen, but instead form "islands" with free space in between. This presented an obstacle for analysis. Our standard images are ~300 μm in the X and Y dimensions, but only 3 of the 35 regions analyzed in this experiment were confluent over that area. To overcome this obstacle we reduced the area of analysis to 60 μm x 60 μm (Fig 4E and 4F). This introduced a bias to the experiment because areas of analysis had to be determined by the experimenter. To limit this bias we restricted the analysis to those density regimes most closely associated with the three organized architectures (S1 Fig). As observed on the uncoated coverslips, Disorganized architectures predominate at densities above 5x10$^3$ cells/mm$^2$ and Immature architectures do not develop. Intermediate architectures can be observed at their expected density regime.

Taken together, our modeling and experimental results support a mechanical explanation for the transition from Immature to Intermediate architecture. They also suggest that densification is the main driver of this transition, whereas cell-substrate adhesion and active cell spreading, a parameter not accounted for in our earlier conceptual model for monolayering, play a less important role.

## Cell-cell adhesion facilitates the transition from immature to intermediate architecture but is not a major factor

MDCK cells A) express the cell-cell adhesion factor E-cadherin, which has long been associated with epithelial identity, and B) are one of the few cultured cell types that effectively epithelialize [47,48]. We therefore investigated the question of how cell-cell adhesion contributes to the transition from Immature to Intermediate architecture. Starting with our standard cell-cell adhesion strength of 0.2 N/m–a value determined empirically for cultured cells expressing E-Cadherin [49]–we modulated the strength of cell-cell adhesions in our simulations by two orders of magnitude in either direction and determined cell-substrate border length and cell-cell border length. We found that while reducing cell-cell adhesion strength does not prevent the development of Intermediate architecture in our simulations, the density at which Immature architecture transitions to Intermediate (dashed line in Fig 5A') increases ($3.1 \times 10^3$ cells/mm$^2$ in our control versus $3.6 \times 10^3$ cells/mm$^2$ when adhesion is reduced to 0.002 N/m) (Fig 5A). These results suggest that cell-cell adhesion facilitates the development of Intermediate cell shapes at lower densities ($< \sim 4 \times 10^3$ cells/mm$^2$), but is not an absolute requirement (Fig 5B).

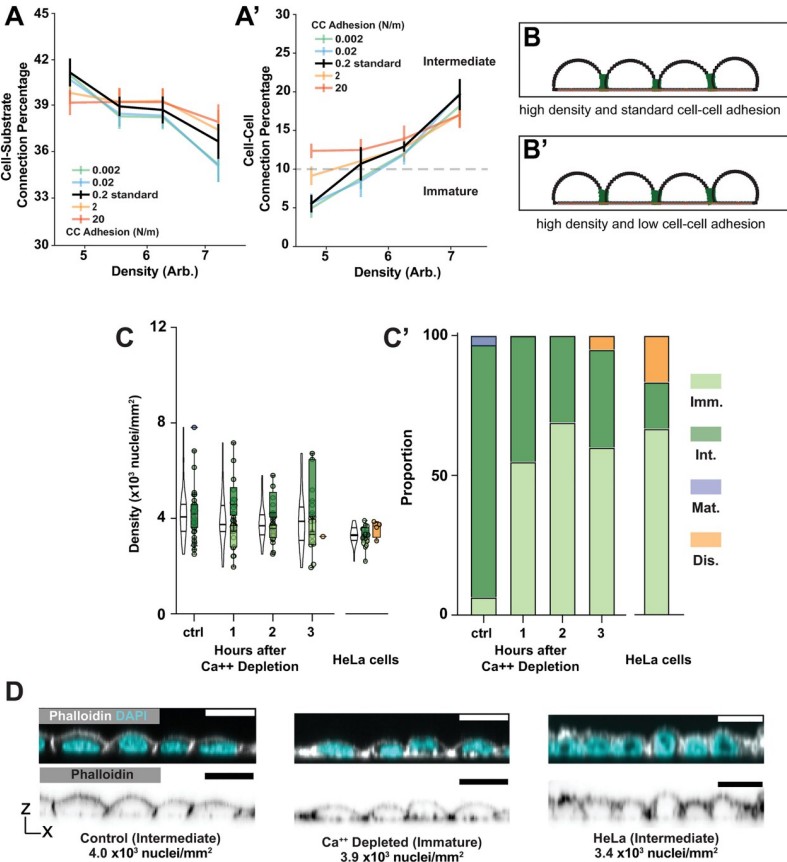

**Fig 5. Intermediate monolayers form with decreased cell-cell adhesion.** A) Cell-substrate connections decrease (A) and cell-cell connections increase (A') as cell density increases across tested values of cell-cell adhesion strengths in simulations. A') The dashed line represents the cell-cell connection percentage required for Intermediate architecture to arise in the model. B) Model representative displaying the final result from decreasing cell-cell adhesion strengths. C, C') Calcium depletion leads to Immature layer architectures in the low density regime where Intermediate architectures are expected. HeLa cells develop Immature, Intermediate and Mature architectures. D) Representative images show differences in morphology between control and calcium depleted layers at comparable densities. Representative HeLa cells shown with an intermediate morphology.

To test this prediction, we allowed MDCK cells to develop Intermediate architectures under standard conditions (200K cells plated for 24Hr in a 1 cm$^2$ culture area), then replaced the culture medium with a calcium-free medium to disrupt calcium-dependent cell adhesion (cadherins) [50]. Although prolonged calcium withdrawal (~6 days) is associated with desmosome-dependent hyper-adhesivity [51] our measurements were performed after only 3 hours and therefore avoided that concern. We find that calcium depletion causes a retrograde shift in layer architecture–from Intermediate "back" to Immature–but only at the lowest densities at which Intermediate architectures develop ($\leq$ 4x10$^3$ nuclei/mm$^2$) (Figs 5C and S1). Calcium depletion appears to diminish but not eliminate tight junctions (as marked by ZO-1 immunostaining) in these cells (S6A Fig).

A potential caveat to interpretating the shift in layer architecture is that calcium depletion might also impact cell shape by reducing non-muscle myosin II activity, and therefore cell contractility [52]. We tested this using Blebbistatin, which inhibits non-muscle myosin II directly. Blebbistatin had no significant impact on architecture, indicating that the effect of calcium depletion is most likely a consequence of reduced cell-cell adhesion (S6B Fig).

As a second approach we examined architecture development in HeLa (Henrietta Lacks) cells, which do not express E-cadherin (S6C Fig) [53]. We cultured these cells to 3.6-4x10$^3$ nuclei/mm$^2$ (their maximum density in our laboratory) using the same collagen-coated culture wells used for our MDCK cell experiments. We suspect that this density does not correspond directly to those observed for MDCK cells because HeLa cells are larger, in which case they should be expected to achieve comparable crowding at a lower cell density [54]. In addition to the expected Immature architectures, we detected Disorganized and, at higher densities, Intermediate architectures (Fig 5C). Together our modelling and experimental work show that adhesion facilitates the transition from Immature to Intermediate architecture at low densities but is not necessary for this transition at higher densities.

## Some intermediates are more equal than others

Our results show that the appearance of lateral cell surfaces in culture–our definition for the transition between Immature and Intermediate–is not restricted to epithelial cells (Fig 5D). This led us to ask the question of how well the Intermediate category recognized by ALAn reflects epithelial architecture. To address this question, we examined the development of another epithelial characteristic, namely cell shape regularity with respect to the tissue surface, as cells densify. Across our large control data set we find that the relationship between density and circularity (shape regularity) is linear up to a circularity of ~0.84, at which point it begins to plateau (Fig 6A).

We find that circularity decreases after calcium depletion, from an average of ~0.84 in the control to ~0.81 at three hours without calcium, and that this is not due to a change in cell density (Fig 6B and 6C'). Consistent with this, HeLa cells at high densities do not achieve the polygonal cell shapes associated with epithelia, but rather retain their characteristic spindle-shaped morphology even when their apical-basal architecture is recognized as Intermediate (S7A Fig). These findings show that shape regularity is not simply proportional to cell density. Instead, shape regularity is proportional to a linear superposition of factors dependent on cell density and cell-cell adhesion. Together, our observations suggest that a uniquely "epithelial" architecture (relying on cadherin-mediated adhesion) corresponds to a shape regularities of ~0.84 and above.

## Subclassification of intermediates: IntA and IntB

We next asked whether layers below and above the 0.84 circularity breakpoint could be distinguished in respect to their apical-basal architecture. Our analysis pipeline, ALAn, discerns the

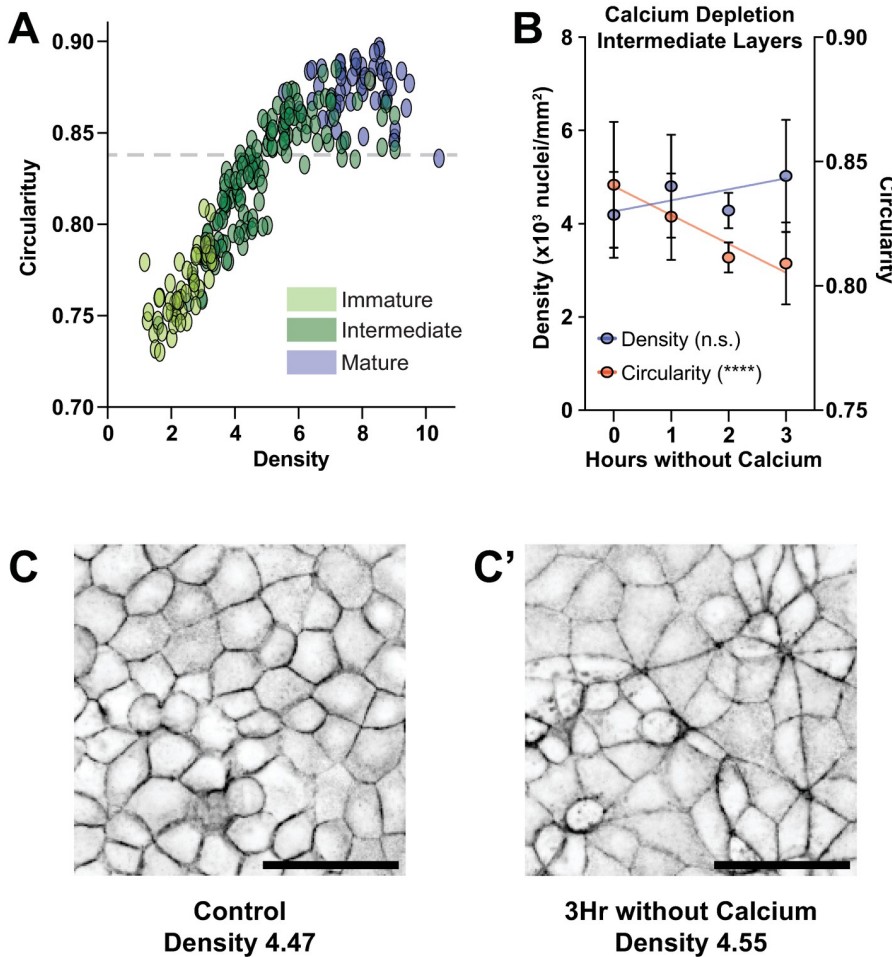

**Fig 6. Intermediate epithelial architecture can be divided into two sub-categories.** A) Density and cell shape regularity (circularity) have a linear relationship until a circularity of approximately ~0.84, at point circularity plateaus. B). Cell shape regularity decreases in MDCK cells as a function of hours without Calcium. The slope has a $p < 0.0001$ to be non-zero. Significance was determined using an F-Test. Density of analyzed layers is unchanged as a function of hours without Calcium. The slope has a $p = 0.0651$ to be non-zero. Significance was determined using an F-Test. C) Representative images showing the change in cell shape (with respect to the tissue surface) after calcium depletion.

shape of lateral and apical surfaces at multi-cell scale using a profile of actin intensity projected along the apical-basal tissue axis (examples Fig 7A–7D). A "shoulder," corresponding in position to lateral actin at cell-cell borders, is evident in the actin intensity profile of a Mature layer (Fig 7A). The basal side (left in the profile) of the shoulder becomes evident as lateral surfaces develop, and the apical side (right in the profile) as apical cell surfaces flatten. The ratio between the maximum slopes on either side of the shoulder, defined here as the Lateral-to-Apical Shape Index, is therefore an indicator of architecture development. Our image analysis pipeline ALAn defines Mature architectures as those in which the Lateral-to-Apical Shape Index is greater than or equal to 1 [8].

ALAN does not use the Lateral-to-Apical Shape Index to distinguish Intermediate architectures from Immature architectures. Rather, these are distinguished by A) an apically-biased asymmetry in the actin intensity plot and B) a change in the relationship between peak actin intensity and peak nuclear distribution across the apical-basal axis, such that the former is apical to the latter (meaning that an apical surface has started to develop in the cultured tissue)

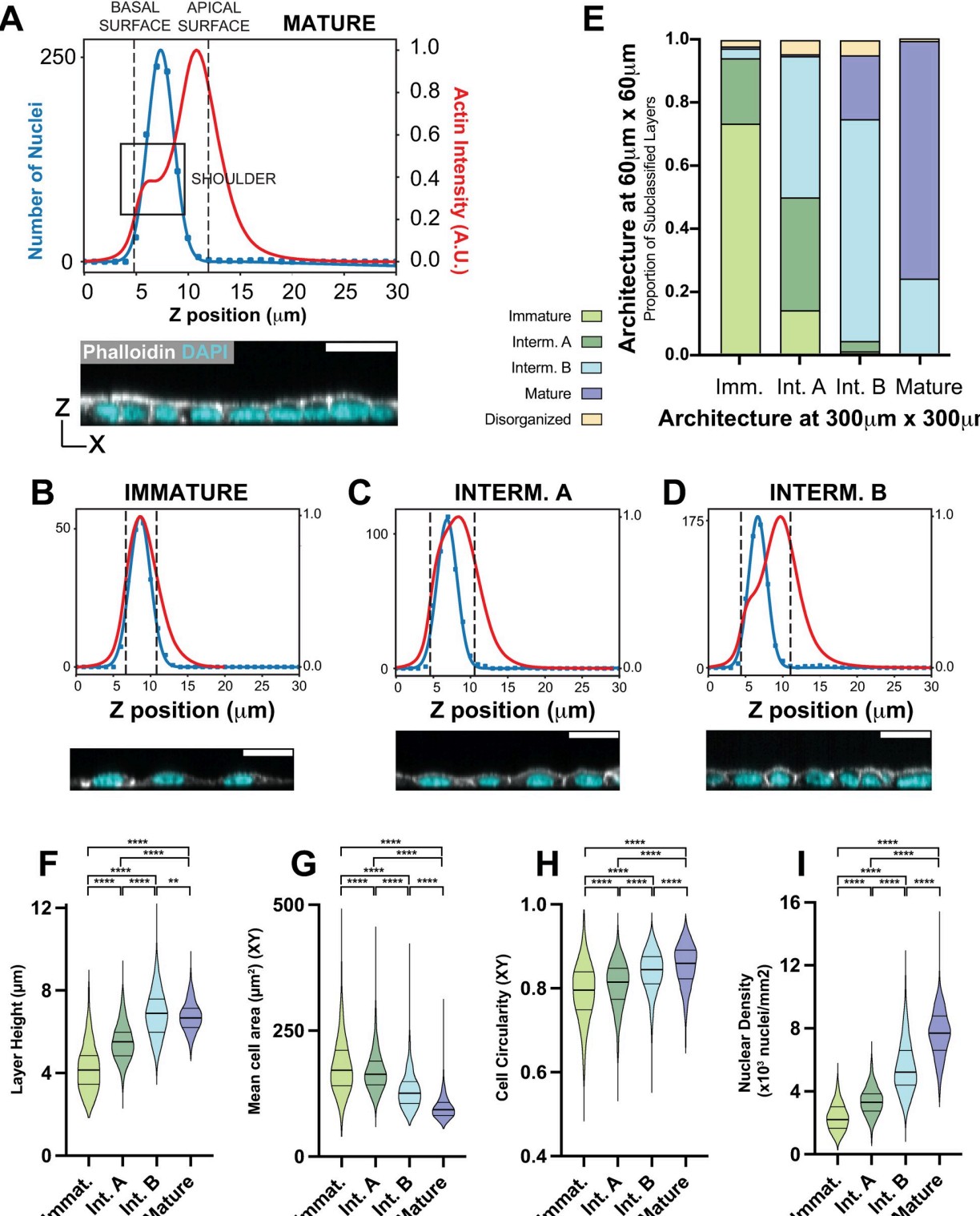

**Fig 7. Subclassified Intermediate A and Intermediate B epithelial architectures are physically distinct.** A) ALAn-generated Nuclei and Actin distribution plots across the apical-basal depth of a tissue layer, formed by projecting Actin onto the z axis, and making a histogram of segmented nuclei positions. A distinct 'shoulder' is present in the plot of actin density against z position in Mature epithelial layer types, as indicated by the box, corresponding to distinct pools of lateral and apical actin. ALAn uses the derivative of the actin intensity plot to detect this shoulder. If the derivative is two-peaked, the ratio of the right peak to the left peak is used. A ratio of greater than 1 is classed as a Mature layer. B,C and D) We now

introduce a further sub-categorization of Intermediate layers based on the projected actin and nuclei profiles of layers across the apical-basal axis. Immature layers are characterized by a lack of defined lateral surfaces. Therefore, in these layers the peak nuclear distribution is located at, or occasionally above, the peak actin intensity (B). Intermediate layers develop a lateral surface and an asymmetric Actin intensity that peaks closer to the surface. Intermediate 'A' layers do not develop a clear shoulder (C). Intermediate 'B' layers develop a clear shoulder with a Lateral to Apical Shape Index of less than 1 (D). E) Proportions of each 60 μm x 60 μm sublayer classification found in organized 300 μm x 300 μm layers. F) Intermediate B sublayers (6.9 μm) are taller than Intermediate A sublayers (5.5 μm). P values from left to right: $p < 0.0001$, $p < 0.0001$, $p < 0.0001$, $p < 0.0001$, $p < 0.0001$, $p = 0.0016$. G) Mean cell area is smaller in Intermediate B sublayers (131 μm$^2$) than Intermediate A sublayers (169 μm$^2$). All $p$ values are $p < 0.0001$. H) Cells are more regular in intermediate B sublayers (0.84) than in Intermediate A sublayers (0.80). All $p$ values are $p < 0.0001$. I) Intermediate B sublayers (5.61x10$^3$ cells/mm$^2$) are denser than Intermediate A sublayers (3.41x10$^3$ cells/mm$^2$). All $p$ values are $p < 0.0001$.

[8]. We examined how this Index evolves in early architectures, and whether it could be used as a means of assessing architecture development within the Intermediate group. We find that in the majority of Intermediate layers (80/141) in a large control data set, the Index cannot be determined. This is because the basal side of the shoulder develops before the apical side, indicating that lateral surface development precedes apical surface flattening. Nearly all (74/80) of these layers have an apical surface circularity below 0.84. Of the remaining 61 Intermediates, 52 are above 0.84. On the basis of these findings, we divided Intermediate apical-basal architectures into two categories, Intermediate A (IntA) and Intermediate B (IntB) (S7B Fig).

We considered whether the distinction between IntA and IntB could have a technical, rather than biological, explanation. ALAn uses a 300 x 300 μM culture area as its standard, but we showed previously that this area can be broken into smaller subregions that may differ in architecture from the larger-scale determination [8]. We therefore asked if the IntB category is a consequence of averaging Mature and Intermediate architectures. One way to address this question is by reducing the area of analysis to 60 x 60 μm (as in Fig 4E and 4F) at which scale averaging effects should be diminished. We find that IntA and IntB can be distinguished even over these smaller regions; in our large control data set, the IntB 300 x 300 μM areas are predominantly comprised of IntB 60 x 60 μm subregions (Fig 7E). The second way to address this question is to examine additional aspects of architecture. With respect to layer height, cross sectional cell area, circularity, and nuclear density, differences are observed at both the larger (300 x 300 μm) and smaller (60 x 60 μm) scales and are consistent between them (Figs 7F–7I and S8A–S8D). Notably, average layer height in IntB regions (and subregions) is greater or equal to the layer height in either IntA or Mature regions (Figs 7A and S8A), meaning that it cannot be explained by averaging between those types.

Although averaging does not explain the IntB category, it does appear to have impacted the IntA category. The 300 x 300 μM areas classed as IntA are comprised of roughly equal proportions of IntA and IntB subregions, with a smaller number of Immature subregions (Fig 7E). Thus at least some of these larger areas are likely to be classified as IntA because they include a mixture of Immature and IntB architectures. We maintain however that the IntA category is genuine because this architecture: 1) can be distinguished at small scale and 2) is morphologically distinct from the Immature category.

Taken together, our results show that the appearance of lateral surfaces (cell-cell borders) in culture is insufficient as an indicator of epithelial morphology. We define a novel morphological transition (IntA to IntB) in densifying MDCK cells that is associated with cadherin-mediated cell-cell adhesion. Our results suggest therefore that this transition reflect the first appearance of a uniquely epithelial architecture in culture.

## Discussion

In this study we investigated the mechanics of epithelial architecture development in culture. Using our previously described extensive dataset and unbiased imaging analysis software, we

identified a developmental series of architectures that appear at characteristic density regimes (S1 Fig) (3). A Mature epithelial architecture is characterized by maximum cell density, polygonal cell arrangement (with respect to the tissue surface) reflected by maximum cell shape regularity, tall cell-cell borders (lateral surfaces), and flat cell apices. Immature architecture is characterized by the minimum density for confluence, irregular cell packing, a lack of height, and a "fried-egg" appearance in which apical-basal cell shape is determined only by the nucleus. Here, through carefully dissecting the contributions of cell density and cell-cell adhesion to layer topology, we find that the transitionary morphologies that connect Immature to Mature can be divided into two distinct phases. The first of these, called Intermediate A (IntA), arises as a simple consequence of cell crowding. While cell-cell adhesion facilitates the onset of Intermediate A architecture, it may not be a strict requirement; even a non-epithelial cell type (HeLa) can exhibit this architecture at high enough density (Fig 5D). The second transitionary phase, Intermediate B (IntB), relies on cell-cell adhesion and we therefore propose that it represents the initial apical-basal architecture that is special to epithelia. Our new developmental regimes and their characteristic shape parameters are described in Fig 8.

The distinction between IntA and IntB architectures corresponds to transitions observed with respect to both the apical-basal axis and the tissue surface. With respect to the former, the onset of IntB corresponds to the initial flattening of the apical surface, as revealed by the plot of actin intensity projected across the depth of the tissue (Fig 7B). With respect to the latter, the onset of IntB corresponds to a plateau in cell shape regularity (Fig 6A). This plateau might be expected of a rheological transition, as individual cell shape changes should be minimal once the tissue is jammed. In agreement with this, the observed plateau corresponds well with a previously determined shape parameter that predicts jamming. This parameter, S0 $S_0 = \sqrt{(4\pi/Circularity)}$ is 3.81 in a 2D vertex model and 3.86 in our system [11]. We speculate that the transitions associated with IntB—apical flattening and fluid to rigid tissue dynamics–are functionally linked. Recent work revealed that MDCK cells follow a 'sizer' behavior; smaller cells will grow more than larger cells during the cell cycle [55]. Sizer behavior would cause cell sizes to tend toward homogeneity, and in turn cell shape regularity. In agreement with this prediction, our work shows that cell shapes (with respect to the tissue plane) become more regular as MDCK cells densify.

In epithelia, contractility and cell-cell adhesion forces are governed largely by an actomyosin belt surrounding each cell. This belt connects with adherens junctions—specialized epithelial junctions that include E-cadherin—near to the apical surface [56]. Adhesion is also provided by another type of epithelial junction, the tight junction, which is located apical to the adherens junction [57]. We showed previously that whereas E-cadherin can be detected in all apical-basal layer architectures, ZO-1, a critical component of the tight junction, is partially observed in Intermediate layers and strongly observed in Mature layers [8]. Disruption of ZO-1 in epithelia can lead to increased contractility along apical cell-cell borders [58].

The question of which mechanical factors drive the IntA-IntB and IntB-Mature transitions remains unanswered. The IntA-IntB plateau in cell shape regularity, corresponding closely to the vertex model-predicted jamming transition, suggests that there are changes in the contractility of cells at this point. Apical contractility, and asymmetric adhesion conferred by adherens and tight junctions, might account for these changes. However, inclusion of these processes in our model requires a more detailed description of short-range repulsion forces to avoid cell-cell overlaps in simulations. This is a goal for future work.

While the regulation of cell polarity—and the impact of polarity on cell shape and tissue organization—have been well-studied in the apical-basal axis, the contribution of cell mechanics in this axis has received less attention. Our novel multinodal 'deformable polygon' model

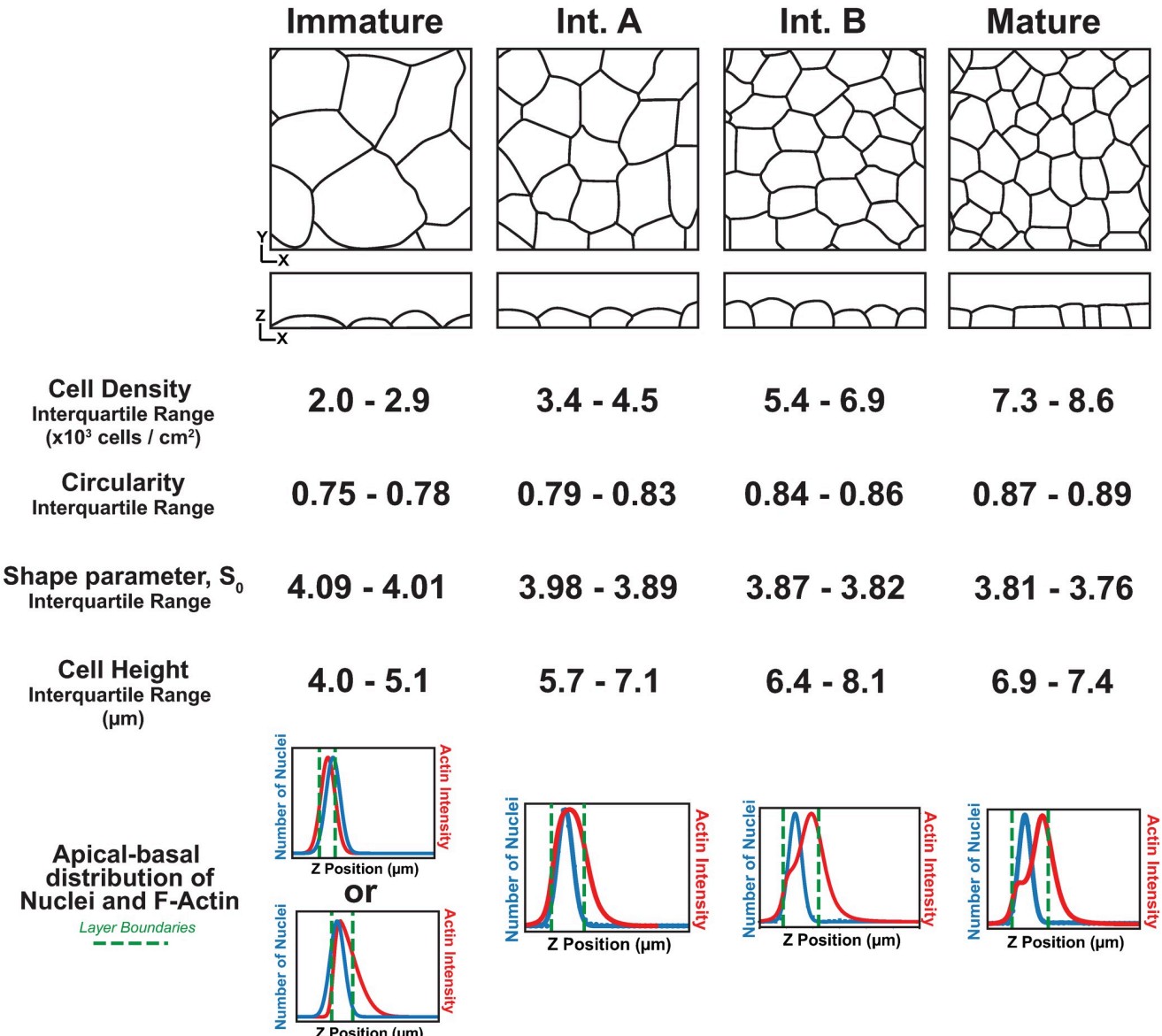

**Fig 8. MDCK cells form a developmental series of architectures in culture.** Cartoons summarizing the distinct epithelial architecture categories made by cultured MDCK epithelial cells along with their defining average density, shape regularity, height and spatial profiles of actin and nuclei across the apical-basal axis.

for apical-basal shape facilitates the study of mechanical contributions in the establishment and maintenance layered topology. We anticipate that this approach can be used to make additional predictions about the physical parameters that govern epithelial architecture.

## Limitations of the study

- Both cell lines used in our study are notoriously heterogenous between labs [59,60]. We derived the "standard" adhesion strengths in our computational model from experimental results in the literature, but cannot be confident that these are accurate to our cell type.

- Calcium depletion will abrogate cadherin-based adhesions. The adhesion complement and organization of real epithelial cells is complex, even in the well-studied Marbin-Darby Canine Kidney (MDCK) cultured epithelial cell model [61]. We do not know the full adhesion profile of the MDCK cells in our lab, or to what extent non-cadherin-based adhesions serve to connect component cells together. (A similar concern applies to our HeLa cells). Additionally, because calcium ions are multipurpose signaling molecules, calcium depletion is likely to impact processes besides cell-cell adhesion [62].

- We do not know the extent to which cell-substrate adhesion in our cells is impacted by the absence of collagen, nor do we know the entire cell-surface receptor profile of our MDCK cells. In agreement with previous work, we observe that cell spreading is diminished in the absence of collagen and more severely on PDMS substrate, but we do not know by how much.

## Materials and methods

### Computational modeling and analysis

Our model considers the movement over time of cell cross-sectional shapes in the XZ plane, including a fixed, rigid substrate coincident with the X axis. Each cell cross-section is represented as a closed curve that is discretized into a varying number of nodes. The substrate is discretized similarly. Cell node positions are updated according to the equation of motion

$$\overrightarrow{F_j^i} = \text{ß} \, \overrightarrow{v_j^i},$$

where $\overrightarrow{F_j^i}$ is the net force on the $j^{\text{th}}$ node of the $i^{\text{th}}$ cell, ß is a viscous drag coefficient, and $\overrightarrow{v_j^i}$ is the velocity of the $j^{\text{th}}$ node of the $i^{\text{th}}$ cell. The net force $\overrightarrow{F_j^i}$ is given by

$$\overrightarrow{F_j^i} = \overrightarrow{F_{internal_j}^i} + \overrightarrow{F_{cs_j}^i} + \overrightarrow{F_{cc_j}^i} + \overrightarrow{F_{spread_j}^i} + \overrightarrow{F_{gravity_j}^i},$$

a sum of force contributions that are defined as follows. The internal regulation of cell size and shape is modeled by each cell tending its cross-sectional area $A$ toward a preferred value $A_0$, while minimizing its cortical perimeter $L$, in the absence of other forces. This is captured by the energy functional

$$E_{int} = \frac{1}{2} k_A (A - A_0)^2 + \frac{1}{2} k_L L^2,$$

where the parameters $k_A$ and $k_L$ control the relative strengths of the area and perimeter constraints. Taking the negative gradient of the above energy with respect to nodal coordinates, we arrive at the internal force on the $j^{\text{th}}$ node of the $i^{\text{th}}$ cell, $\overrightarrow{F_{internal_j}^i} = (F_x, F_z)$, where

$$F_x = -\frac{1}{2} k_A (A - A_0) \frac{\sum_k \left( x_k z_{k+1} - x_{k+1} z_k \right)}{\left| \sum_k \left( x_k z_{k+1} - x_{k+1} z_k \right) \right|} \left( z_{j+1} - z_{j-1} \right)$$

$$- k_L L \left( \frac{x_j - x_{j-1}}{\sqrt{\left( x_j - x_{j-1} \right)^2 + \left( z_j - z_{j-1} \right)^2}} - \frac{x_{j+1} - x_j}{\sqrt{\left( x_{j+1} - x_j \right)^2 + \left( z_{j+1} - z_j \right)^2}} \right),$$

$$F_z = \frac{1}{2}k_A(A - A_0)\frac{\sum_k (x_k z_{k+1} - x_{k+1}z_k)}{|\sum_k (x_k z_{k+1} - x_{k+1}z_k)|}\left(x_{j+1} - x_{j-1}\right)$$

$$- k_L L\left(\frac{z_j - z_{j-1}}{\sqrt{\left(x_j - x_{j-1}\right)^2 + (z_j - z_{j-1})^2}} - \frac{z_{j+1} - z_j}{\sqrt{\left(x_{j+1} - x_j\right)^2 + (z_{j+1} - z_j)^2}}\right).$$

External adhesion is modeled by linear springs that link cell or substrate nodes when they come into proximity. A spring breaks when connected nodes move outside of a threshold interaction distance. The force on the $j^{\text{th}}$ node of the $i^{\text{th}}$ cell due to adhesion to the substrate is given by

$$\overrightarrow{F_{csj}^i} = \begin{cases} -\gamma_{cs}\sum_k \left(\left|\overrightarrow{p^i}_j - \overrightarrow{s_k}\right| - l_{cs}\right)\frac{\overrightarrow{p^i}_j - \overrightarrow{s_k}}{\left|\overrightarrow{p^i}_j - \overrightarrow{s_k}\right|} & \text{for } \left|\overrightarrow{p^i}_j - \overrightarrow{s_k}\right| < d_{cs} \\ 0 & \text{else} \end{cases},$$

where $\overrightarrow{s_k}$ is the $k$th substrate node position, $\overrightarrow{p^i}_j$ is the position of the $j^{\text{th}}$ node of the $i^{\text{th}}$ cell, $\gamma_{sub}$ controls the adhesion strength between cells and the substrate, $l_{sub}$ is the natural length of a cell substrate adhesion, and $d_{cs}$ is the maximum interaction distance for a substrate adhesion. We enforce that substrate nodes can connect only to the closest cell node, and each cell node can connect only to the closest substrate node. Similarly, the force on the $j^{\text{th}}$ node of the $i^{\text{th}}$ cell due to cell-cell adhesion is given by

$$\overrightarrow{F_{ccj}^i} = \begin{cases} -\gamma_{cc}\sum_l\sum_k \left(\left|\overrightarrow{p^i}_j - \overrightarrow{p^l}_k\right|^2 - l_{cc}\right)\frac{\overrightarrow{p^i}_j - \overrightarrow{p^l}_k}{\left|\overrightarrow{p^i}_j - \overrightarrow{p^l}_k\right|^2} & \text{for } l_{cc} < \left|\overrightarrow{p^i}_j - \overrightarrow{p^l}_k\right|^2 < d_{cc} \\ -R_{cc}\sum_l\sum_k \left(\left|\overrightarrow{p^i}_j - \overrightarrow{p^l}_k\right|^2 - l_{cc}\right)\frac{\overrightarrow{p^i}_j - \overrightarrow{p^l}_k}{\left|\overrightarrow{p^i}_j - \overrightarrow{p^l}_k\right|^2} & \text{for } \left|\overrightarrow{p^i}_j - \overrightarrow{p^l}_k\right| < l_{cc} \\ 0 & \text{else } 0 \end{cases},$$

where $\overrightarrow{p^l}_k$ is the position of the $k^{\text{th}}$ node of the $l^{\text{th}}$ cell, $\gamma_{cc}$ controls the adhesion strength between cells, $l_{cc}$ is the natural length of a cell-cell adhesion, and $d_{cc}$ is the maximum interaction distance for a cell-cell adhesion. When $\left|\overrightarrow{p^j_i} - \overrightarrow{p^k_l}\right|^2 < l_{cc}$, we replace $\gamma_{cc}$ with a constant repulsive strength $R_{cc}$, since cell repulsion should be independent of cell-cell adhesion. This allows for us to test arbitrarily small cell-cell adhesion strengths.

The gravitational and spreading forces are given by

$$\overrightarrow{F_{gravity}^i}_j = \begin{cases} C_G\langle 0, -1\rangle & \text{for all nodes in cells without a path to the substrate} \\ 0 & \text{else} \end{cases}$$

$$\overrightarrow{F_{spread}^i}_j = \begin{cases} C_s\left\langle \pm\frac{1}{\sqrt{2}}, \frac{1}{\sqrt{2}}\right\rangle & \text{for all } j \text{ adjacent to the outermost substrate connections with CIL} \\ 0 & \text{else} \end{cases}$$

where $C_G$ is a constant parameter, and $C_s$ can depend on $\gamma_{cs}$. $C_s$ takes the following form

depending on the model used to simulated active spreading

$$
C_s = \begin{cases}
C_s & \text{constant scaling} \\[2ex]
C_s\left(\dfrac{\gamma_{cs}}{\gamma_{cs,0}}\right) & \text{linear scaling} \\[2ex]
C_s\left(\dfrac{\gamma_{cs}}{\gamma_{cs,0}}\right)^{0.2} & \text{nonlinear scaling}
\end{cases}
$$

where $\gamma_{cs,0}$ is the reference cell-substrate adhesion described in Table 1.

Gravity is estimated to be four to six orders of magnitude smaller than the other forces in this system, and is therefore excluded from consideration once cells reach the substrate or a cell connected to the substrate, as expected in a low Reynolds number environment [63].

Spreading is described in the main text. Briefly, the nodes nearest to the outermost substrate connections will spread out from the cell, and down toward the substrate provided there are no cell-cell contacts on the side in question. To set the nonlinear scaling between cell-substrate adhesion and cell spreading force described above, we used a power law. In our nonlinear scaling, we chose the spreading force to be proportional to the adhesive force fraction of normal to the one fifth.

The feedback mechanism is described in the main text. Briefly, cells in contact with one another undergo changes in cortical actin polymerization and cortical remodeling. The feedback mechanism operates as follows:

$$
\gamma_{cc} = \begin{cases}
\gamma_{cc} & \text{Without cell} - \text{cell contact} \\[1.5ex]
\gamma_{cc}(1 + 0.03 N_{sub}) & \text{With cell} - \text{cell contact}
\end{cases},
$$

$$
\gamma_{cs} = \begin{cases}
\gamma_{cs} & \text{Without cell} - \text{cell contact} \\[1.5ex]
1.5\gamma_{cs} & \text{With cell} - \text{cell contact}
\end{cases}
$$

**Table 1. Reagents used in this study.**

| Reagent | Supplier | Product Code |
| --- | --- | --- |
| Uncoated 8-well μ-slide | Ibidi | 80821 |
| Collagen IV coated 8-well μ-slide | Ibidi | 80822 |
| 0.25% Trypsin-EDTA (1x) | Gibco | 210200–056 |
| Trypan Blue Stain 0.4% | Gibco | 15250–61 |
| Rabbit anti-Ecadherin | Cell signaling | 24E10 |
| Blebbistatin, 50 μM | Sigma | B0560 |
| Alexa Fluor 633 Goat anti-Rabbit | Invitrogen | A21071 |
| Alexa Fluor 633 Goat anti-Rat | Invitrogen | A21094 |
| Fluorescein Phalloidin | Invitrogen | F432 |
| Vectashield antifade mounting medium with DAPI | Vector Laboratories | H-1200 |
| CellMask Orange | Invitrogen | REFC10045 |
| DMEM/F12(1:1) (1x) | Gibco | 11330–032 |
| Penicillin-Streptomycin (10,000 U/mL) | Gibco | 15140–122 |
| Fetal Bovine Serum | Gibco | 26140079 |
| SYLGARD 184 Silicone Elastomer | World Precision Instruments | SYLG184 |
| Silicone PDMS Membrane 0.005" Thick | Interstate Specialty Products | SSPM823-005 |

$$d_{cc} = \begin{cases} d_{cc} & \text{Without cell} - \text{cell contact} \\ 1.2d_{cc} & \text{With cell} - \text{cell contact} \end{cases}$$

where $N_{sub}$ is current number of cell-substrate connections for the cell.

Cells remodel their cortex; if the spacing between adjacent nodes exceeds twice the average spacing, a new node is added at the bisection. If the spacing between adjacent nodes becomes smaller than half the expected length, the node which is closest to its next nearest neighbor will be removed. Remodeling in this way keeps the spatial discretization of cell shapes at an approximately uniform density throughout a simulation. Cells nodes tend to spread greater than their average separation without remodeling and will prevent accurate representations of cell cortices.

The cell node equations of motion are integrated numerically using an explicit Euler method timestep Δt whose value is chosen to ensure numerical stability. The value of Δt is set to 1. We found that in our model ~750 timesteps correspond to one hour in culture by comparison of the spreading phenotype observed in a pair of adjacent cells (S2 and S3 Movies). At each timestep remodeling is applied before the calculation of forces and the moving of nodes. We noticed no morphological differences between simulations as a function of the average number of nodes in a cell.

Cell division was implemented by having cells double in size and cease cell-cell adhesions over the final hour of their cell cycle. We calibrated the time scale for our model using the time it takes for two cells to spread. Division occurs after growth in one time step and the daughter cells undergo multiple rounds of cortical remodeling to populate the cytokinetic plane with nodes. The division axis is determined by the topmost and bottommost (in z) cell nodes to create planar divisions and produce two daughter cells. The cell cycle duration for each initial cell is sampled independently from a random uniform distribution $U[0, 4]$ hours.

The dimensional and non-dimensionalized parameter values used are provided in Table 2. These values informed by our experimental estimate of ~100 μm$^2$ for the typical cross-sectional area of MDCK cells in culture, and reported values of physical forces for cultured cells [42,49,64].

Computational modeling was undertaken using Python 3.8.3 in a Jupyter notebook. Packages used for this model are: Numpy (https://numpy.org/), Matplotlib (https://matplotlib.org/), and NetworkX (https://networkx.org/). Automated Layer Analysis (ALAn) was used to

**Table 2. Model parameter values used in this study.**

| Parameter | Dimensional Value | Non-dimensionalized Value | Source |
|---|---|---|---|
| Preferred Area, $A_0$ | ~100 μm$^2$ | 1 | Observation |
| 2D Area Stiffness, $k_A$ | ~2e7 N/m$^2$ | 1 | Calculated, [67](by analogy) |
| Cortical Stiffness, $k_L$ | ~0.1 N/m | 0.0005 | [64] |
| Cell-Cell Adhesion Strength, $\gamma_{cc}$ | ~0.2 N/m | 0.001 | [49] |
| Cell-Cell Repulsion Strength, $R_{cc}$ | ~10 N/m | 0.05 | Estimate |
| Cell Substrate Adhesion Strength, $\gamma_{cs}$ | ~20 N/m | 0.1 | [42] |
| Cell-Cell Interaction Distance, $d_{cc}$ | ~1.6 μm | 0.16 | Estimate |
| Cell-Substrate Interaction Distance, $d_{cs}$ | ~1 μm | 0.1 | Estimate |
| Cell-Cell Adhesion rest length, $l_{cc}$ | ~50 nm | 0.005 | Estimate |
| Cell-Substrate Adhesion rest length, $l_{cs}$ | ~50 nm | 0.005 | Estimate |
| Spreading Strength, $C_s$ | ~16 μN | 0.008 | Estimate |
| Gravitational Strength, $C_G$ | ~2 μN | 0.001 | Estimate |

characterize cultured layer architecture [8]. A detailed method for the use of ALAn is published [65]. All code for the modelling is available on the Finegan-Bergstralh-Lab GitHub repository: https://github.com/Bergstralh-Lab.

## Apical to basal length ratio

For both simulated and MDCK cells, the apical and basal lengths were found by measuring the line distance along the apical and basal surfaces of cells in an XZ plane. The central XZ plane was chosen for MDCK cells. The surfaces were defined by exclusion from cell-cell borders in both cases.

## Cell culture

MDCK cells were gifted by Patrick Oakes (Loyola University) and originated from the laboratory of W. James Nelson (Stanford) RRID: CVCL_0422. MDCK cells are notoriously heterogenous, and we used cells that exhibit the Type-2 morphology as defined in [66]. HeLa S3 cervical carcinoma cells were gifted by Dragony Fu (University of Rochester). Cells were cultured using standard methods in DMEM supplemented with 10% FBS and 1% Pen/Strep. For standard monolayer culture experiments, cells were passaged one day prior to seeding on 8 well Ibidi chamber slides (uncoated or collagen coated). Media was changed every 24 hrs.

For culture on PDMS membranes, we first cut out a strip of 0.005" thick PDMS membrane the size of a coverslip. Membranes were dipped in methanol to reduce static and placed on a coverslip to allow to dry. A 1 cm by 1 cm culture area was made by curing Sylgard 184 in a petri dish and cutting a thick walled barrier to mimic the Ibidi culture wells. Slides were kept in a sterile plastic tub and allowed to culture for the appropriate time frame.

To grow MDCK cell colonies, cells were plated at an extremely low density of ~50 cells onto a 1 cm$^2$ collagen coated Ibidi culture slide to get isolated seed cells. Cell culture media was changed every two days, and cells were left for 6 days. Colonies like the one shown in Fig 5 were present in roughly 25% of culture wells, while colonies growing at the edge of the growth area were most common.

## Fixation and immunostaining

Cells were washed with dPBS and fixed using ~4% formaldehyde, 2% PBS-tween for 10 minutes. Three washes (10 minutes each) in PBS-0.2% Tween were carried out between fixation and staining.

Primary and secondary antibodies were added at 1:500 dilution. FITC phalloidin was used to stain actin. Vectashield plus DAPI was added directly to the wells. For live imaging, cell mask was added at a concentration of 1:1000 just before imaging. HeLa cells were cultured the same as MDCK cells, but were left for up to a week to promote a homeostatic density.

## Calcium depletion and blebbistatin treatment

Cells were plated and grown as previously described to produce primarily Intermediate architectures [8]. After 24 hrs, media was removed and cells were washed with PBS, after which cells were incubated for the specified time in magnesium-calcium-free PBS supplemented with 2 mM Magnesium dichloride. We found that for treatments of up to 3 hours in calcium free media, cell monolayers remained intact and mostly confluent, though there were occasionally cell scale gaps that would develop in the monolayer. These small gaps are unlikely to affect classification by ALAn and are unlikely to impact average architecture of the tissue.

For blebbistatin treated monolayers, cells were grown to an Intermediate architecture and treated with 50 µg/ml of blebbistatin or the equivalent amount of DMSO as a control for the specified time. After treatments cells were fixed and stained as per the above protocol.

### Imaging

For live cell imaging, we used a Leica SP5 Confocal microscope using a 40x/1.25 HCX PL APO oil objective. Confocal stacks were taken at 10-minute intervals.

Fixed tissue was imaged using an Andor Dragonfly 200 Spinning Disk Confocal microscope with a 40x/1.15 water objective. Confocal stacks were taken beginning beneath the bottom of the chamber slide where no actin signal could be detected and ending above the layer, when no more actin signal could be detected, taken at a z-spacing of 0.23 µm. Images were segmented in Imaris, and the corresponding image and.csv files containing nuclear positional information were used with ALAn as previously described. ALAn was used for all architecture classification. Cells cultured on PDMS membranes did not grow to confluence, so regions of 100 pixels x 100 pixels (~60 µm x 60 µm) were analyzed using ALAn instead of the entire field of view.

To image cell colonies we used a Leica M165 FC Stereomicroscope with a 1.0X objective and a K3M camera.

### Supporting information

**S1 Fig. MDCK cells take on distinct organized architectures.** Representative confocal micrographs summarizing the distinct epithelial architecture categories made by cultured MDCK epithelial cells along with their defining average density, shape regularity in the apical plane, and height values.
(PDF)

**S2 Fig. Active spreading is required in the model to recapitulate live cell behavior.** A) A representative timepoint from live confocal imaging of MDCK cells stained with CellMask upon initial culture plating shows that single cells dropped onto a substrate spread. Scale bar = 20 µm. B and C) Images of the stable shapes formed by single simulated cells without (B) and with (C) active spreading implemented.
(PDF)

**S3 Fig. The computational model does not predict the emergence of a Mature architecture.** The ratio of the apical surface length to the basal surface length of simulated MDCK cells (left) grows as a function of density. Dashed line indicates the density transition between Intermediate and Mature architectures. The same ratio for modeled cells (right). Dashed line indicates predicted transition between Intermediate and Mature architectures.
(PDF)

**S4 Fig. MDCK Colonies densify at the center.** A confocal micrograph showing the distribution of nuclei (fixed DNA staining) in a colony of cultured MDCK cells. This image, also in Fig 4B, is shown here in a heatmap lookup table to emphasize the gradient of cell density starting from the colony center.
(PDF)

**S5 Fig. Cell-substrate adhesion strength affects epithelial architecture.** A) Cells grown on collagen spread to cover area of the substrate that is ~5 times larger cells than cells grown on an uncoated substrate. Representative confocal micrographs z-stacks of cells stained with Cell-Mask grown on Collagen IV and uncoated glass coverslips. Scale bar = 50 µm. B) Cell-

substrate and C) cell-cell contact lengths of cultured MDCK cells decrease and increase respectively as layers transition from immature and intermediate architectures. D-G) Reducing cell-substrate adhesion strength reduces the number of cell-substrate connections in simulations with different implementations of cell spreading force. D) Cell-substrate connections decrease and D') Cell-cell connections increase as a function of density at all substrate adhesion strengths with a constant spreading force. The dashed line (D') represents the connection percentage required for Intermediate architectures to arise in the model. E) Final, stable equilibrium state of simulations with standard (E) and low (E') cell-substrate adhesion. F) Cell-substrate connections decrease and F') Cell-cell connections increase as a function of density at all substrate adhesion strengths when cell spreading force is scaled linearly with substrate adhesion. The dashed line (F') represents the connection percentage required for Intermediate architectures to arise in the model. G) Final, stable equilibrium state of simulations with standard (G) and low (G') cell-substrate adhesion. H) Cell shape regularity (with respect to the tissue surface) of cultured cells is not impacted by the presence of collagen on the substrate. Only Intermediate layers are shown. $p = 0.4692$, Significance was determined using an unpaired two-tailed Student's t test.
(PDF)

**S6 Fig. Cell-cell adhesion facilitates but is not necessary for Intermediate architectures.** Confocal light imaging of immunostained, fixed cells shows that A) ZO-1 localizes to cell-cell junctions apically (green arrows) in mature cultured MDCK cells. Scale bars = 10 μm. After calcium depletion for 3 hrs, cells form a domed morphology. Weak ZO-1 immunoreactivity (green arrows) is evident in these cells. B) Blebbistatin does not affect the presence Intermediate architectures at the densities tested. C) MDCK cells express E-cadherin at cell-cell borders, while E-cadherin is not expressed in HeLa cells. Scale bars = 10 μm. D) Cell shape regularity (with respect to the tissue surface) is not impacted by Blebbistatin treatment. Only Intermediate architectures are shown. $p$ values left to right: $p = 0.1495$, $p = 0.7866$, $p = 0.8101$. Significance was determined using an unpaired two-tailed Student's t test.
(PDF)

**S7 Fig. HeLa cells do not pack regularly in the plane of the epithelium. MDCK cells become regular at the Intermediate A- Intermediate B transition**. A) HeLa cells exhibit spindle-like cell morphologies even in layers classed as Intermediate by our image analysis pipeline ALAn. Representative image. Scale bar = 20 μm. B) MDCK cell shape regularity (with respect to the tissue surface) is significantly lower in Intermediate A architectures in comparison to Intermediate B architectures. $p < 0.0001$. Significance was determined using an unpaired two-tailed Student's t test.
(PDF)

**S8 Fig. Intermediate A and Intermediate B architectures are physically distinct.** A) Intermediate B layers (7.3 μm) are taller than Intermediate A layers (6.3 μm). P values from left to right: $p < 0.0001$, $p < 0.0001$, $p < 0.0001$, $p < 0.0001$, $p < 0.0001$, $p = 0.4497$. B) Mean cell area is smaller in Intermediate B layers (126 μm$^2$) than Intermediate A layers (173 μm$^2$). All $p$ values are $p < 0.0001$. C) Cells are more regular in intermediate B layers (0.85) than in Intermediate A layers (0.81). All $p$ values are $p < 0.0001$. D) Intermediate B layers (6.16x10$^3$ cells/mm$^2$) are denser than Intermediate A layers (3.93x10$^3$ cells/mm$^2$). All $p$ values are $p < 0.0001$. E) Intermediate B sublayers (6.5 μm) drawn from Intermediate A full layers are taller than Intermediate A sublayers (5.7 μm) drawn from the same population. P values from left to right: $p < 0.0001$, $p < 0.0001$, $p = 0.3332$. F) Intermediate B sublayers (7.1 μm) drawn from Intermediate B full layers are taller than Intermediate A sublayers (5.8 μm) drawn from the same

population. $p$ values from left to right: $p = 0.4664$, $p < 0.0001$, $p < 0.0001$.
(PDF)

**S1 Movie. Cell contacts promote cortical activity.** Movie of MDCK cells plated sparsely on a Fibronectin coated substrate. Cells making cell-cell contacts move more than unpaired cells. Movie taken at 10 minutes / frame shown at 10 frames / second. Scale bar represents 50 μm.
(AVI)

**S2 Movie. Isolated MDCK cells landing on substrate.** A live XZ reconstruction of cells landing on a Collagen coated substrate. These cells spread instead of developing a significant cell-cell border. Movie taken at 10 minutes / frame shown at 10 frames / second. Scale bar represents 20 μm.
(AVI)

**S3 Movie. Two cells simulated on unbound substrate.** Two modeled cells are allowed to fall on an unbound substrate in proximity. The modeled cells spread instead of developing a significant cell-cell border. Movie shown at 500 timepoints / second.
(MOV)

**S4 Movie. Four cells simulated on an unbound substrate.** Four modeled cells are allowed to fall on an unbound substrate in proximity. The modeled cells spread instead of developing a significant cell-cell border. Movie shown at 500 timepoints / second.
(MOV)

**S5 Movie. Four cells simulated on a limited substrate.** Four modeled cells are allowed to fall on an unbound substrate in proximity. The modeled cells cannot spread off the substrate and develop a significant cell-cell border. Movie shown at 500 timepoints / second.
(MOV)

**S6 Movie. Cell division in the model.** Three modeled cells are allowed to fall on an unbound substrate in proximity and the middle cell is allowed to divide. After division, one of the daughter cells develops a significant cell-cell border with two of its neighbors. Movie shown at 500 timepoints / second.
(MOV)

## Acknowledgments

We are grateful to the labs of Mark Peifer, Scott Williams, and Holly Lovegrove; the University of Rochester Invertebrate Group; and members of the Bergstralh lab for their questions and comments.

## Author Contributions

**Conceptualization:** Christian Cammarota, Alexander G. Fletcher, Dan T. Bergstralh.

**Data curation:** Christian Cammarota.

**Formal analysis:** Christian Cammarota, Dan T. Bergstralh.

**Funding acquisition:** Dan T. Bergstralh.

**Investigation:** Christian Cammarota, Nicole S. Dawney, Philip M. Bellomio, Maren Jüng, Dan T. Bergstralh.

**Methodology:** Christian Cammarota, Nicole S. Dawney, Alexander G. Fletcher, Dan T. Bergstralh.

**Project administration:** Dan T. Bergstralh.

**Software:** Christian Cammarota, Maren Jüng, Alexander G. Fletcher.

**Supervision:** Alexander G. Fletcher, Dan T. Bergstralh.

**Validation:** Christian Cammarota.

**Visualization:** Christian Cammarota, Tara M. Finegan.

**Writing – original draft:** Christian Cammarota, Tara M. Finegan, Dan T. Bergstralh.

**Writing – review & editing:** Christian Cammarota, Alexander G. Fletcher, Tara M. Finegan, Dan T. Bergstralh.

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
