## [Decision Letter · Decision Letter 0]

12 Jan 2024

Dear Dr Bergstralh,

Thank you very much for submitting your manuscript "The mechanical influence of densification on epithelial architecture" for consideration at PLOS Computational Biology. As with all papers reviewed by the journal, your manuscript was reviewed by members of the editorial board and by several independent reviewers. The reviewers appreciated the attention to an important topic. Based on the reviews, we are likely to accept this manuscript for publication, providing that you modify the manuscript according to the review recommendations.

Sincerely,

Andrea Ciliberto

Academic Editor

PLOS Computational Biology

Daniel Beard

Section Editor

PLOS Computational Biology

Reviewer's Responses to Questions

**Comments to the Authors:**

Reviewer #1: The authors present a cell shape simulator that recapitulates several salient features cell architecture during assembly and maturation of an epithelial sheet. The authors integrate this model with experimental data collected from MDCK cell cultures over a range of densities and under specific confinement. Apical-basal cell shape changes, including contact areas with neighboring cells are generated according to parameters representing biophysical properties of the cell, including cell-cell and cell-substrate adhesion energies. Cell culture experiments are carried out to modulate several of these parameters and compare the results with analogous model perturbations. Overall, the model is interesting and has considerable potential in future applications. The physical role of confinement seems a prerequisite for the formation of a simple single-layered cell sheet modeled by MDCK and even HeLa cells with limited roles for cell-cell adhesion. This later point is surprising, but may be reflect low cell density initial conditions of their model. Future improvements on this model may be able to incorporate epithelial polarity that might enable the model to generate fully mature cell architectures. Overall, the manuscript is well written but contains several oversimplifications of the cell biology of adhesion and the role of substrate adhesion in establishment of polarity.

Some concerns:

+ There are no references to prior simulation studies on the mechanics of epithelial sheets in the transverse dimension. Classical paper by Odell is missing as are multiple Honda and Brodland papers. More recent work by the Munro lab, especially Sherrard et al (2010) should be discussed. While they do not explicitly deal with confinement, polarity and apical-basal cell shape are represented.

+ On page 4, the authors state that "Epithelia most commonly form a monolayered architecture"... Greater care should be taken to properly place the current model into the class of "simple epithelia". There is considerable diversity in epithelial architectures (see Bragulla and Homberg, 2009).

+ I enjoyed the "model building" introduction starting with single cells on an adhesive substrate. However, the immediate inclusion of a phenomenological feedback circuit between cell-cell and cell-substrate adhesion was premature. I would rather this have been introduced to "fix" some deficit. Can this feedback be removed?

+ Several additional phenomenological rules are added to the model, including contact-inhibition-of-locomotion (CIL). Formally, there is no real locomotion. Can this rule be removed? and how does it change the outcome? Do MDCK cells demonstrate CIL in culture? Can this version of CIL emerge just from competition for adhesive substrates?

+ The terms for the various architectures, disordered, immature, intermediate, and mature, imply a progression akin to a mesenchymal to epithelial transition during morphogenesis. However, MDCK cells are an immortalized epithelial cell line and have little to no developmental change. Freshly plated cells already express all the proteins to generate a polarized mature epithelium, there is no progression. Model cells do not have the ability to polarize in this manner and are more representative of non-epithelial cells, such as the HeLa cells used briefly in Fig 5.

+ Could the authors clarify the connection between confinement and density? It seems that density is the product of confinement, and growth, division, or motility. Can density be achieved without this?

+ On page 10, the authors state that the margin of an epithelial patch is "pushed out" by proliferative pressure. Wouldn't there be a "free-edge" response at the margin that would initiate outward directed cell traction? Such traction would necessarily add to the tension across the dorsal surface of the cell sheet and "smooth" the surface.

+ The authors use uncoated wells as a way to reduce substrate adhesion. However, MDCK cells are well known to synthesize their own ECM over time (Yu et al, MBoC 2004). Furthermore, the 10% FBS culture media contains sufficient fibronectin and vitronectin for the cells to assemble more ECM (Hayman et al, 1985). Would it be possible to increase substrate adhesion by providing cells with additional soluble ECM?

+ Calcium withdrawal exposes hyper-adhesivity in MDCK cells. This requires the formation of desmosomes. See Conway group or Garrod group's work.

Minor issues:

+ As a note: it would be helpful if the authors could put their results in the context of recent paper from the Gardel group (Devany et al, 2023). The difference in cell shapes may be related to changes in the cell cycle.

+ On page 14, Yeh-Shiu et al, 2004 is referenced without an inline cite.

Reviewer #2: See attachment.

**Have the authors made all data and (if applicable) computational code underlying the findings in their manuscript fully available?**

Reviewer #1: Yes

Reviewer #2: **No: **It is not clear to me where the code for the models presented here will be available.

PLOS authors have the option to publish the peer review history of their article (what does this mean?). If published, this will include your full peer review and any attached files.

Reviewer #1: No

Reviewer #2: No

Figure Files:

Data Requirements:

Reproducibility:

References:

---

## [Decision Letter · Decision Letter 1]

14 Mar 2024

Dear Dr. Bergstralh,

We are pleased to inform you that your manuscript 'The mechanical influence of densification on epithelial architecture' has been provisionally accepted for publication in PLOS Computational Biology.  Following the second round of revision, we ask you, as suggested by Reviewer 1, to replace the 'simple monolayer' sentence with "a common epithelial architecture".

Best regards,

Andrea Ciliberto

Academic Editor

PLOS Computational Biology

Daniel Beard

Section Editor

PLOS Computational Biology

Reviewer's Responses to Questions

**Comments to the Authors:**

Reviewer #1: I have one small comment that I would appreciate be fixed. The authors state that the "most common epithelial architecture is a 'simple' monolayer." This cannot be demonstrated with any level of confidence. The entire outer epithelial surface of terrestrial vertebrates exhibit a non-simple architecture. Leaving any bias behind I would suggest they said "... a common epithelial architecture...".

Reviewer #2: I thank the authors for having fully answered all my concerns/questions.

**Have the authors made all data and (if applicable) computational code underlying the findings in their manuscript fully available?**

Reviewer #1: Yes

Reviewer #2: Yes

PLOS authors have the option to publish the peer review history of their article (what does this mean?). If published, this will include your full peer review and any attached files.

Reviewer #1: No

Reviewer #2: No

---

## [Editor Report · Acceptance letter]

25 Mar 2024

PCOMPBIOL-D-23-01699R1 

The mechanical influence of densification on epithelial architecture

Dear Dr Bergstralh,

I am pleased to inform you that your manuscript has been formally accepted for publication in PLOS Computational Biology. Your manuscript is now with our production department and you will be notified of the publication date in due course.

With kind regards,

Anita Estes
